# Transformation of Combustion Nanocatalysts inside Solid Rocket Motor under Various Pressures

**DOI:** 10.3390/nano9030381

**Published:** 2019-03-06

**Authors:** Jun-Qiang Li, Linlin Liu, Xiaolong Fu, Deyun Tang, Yin Wang, Songqi Hu, Qi-Long Yan

**Affiliations:** 1Xi’an Modern Chemistry Research Institute, Xi’an 710065, China; llijq@sohu.com (J.-Q.L.); fuxiaolong204@163.com (X.F.); 2Science and Technology on Combustion, Internal Flow and Thermo-structure Laboratory, Northwestern Polytechnical University, Xi’an 710072, China; lll@nwpu.edu.cn (L.L.); tangdy@mail.nwpu.edu.cn (D.T.); wongyin@mail.nwpu.edu.cn (Y.W.); pinecore@nwpu.edu.cn (S.H.)

**Keywords:** solid propellants, condensed products, catalytic combustion, compositions, rocket motor

## Abstract

In this paper, the dependences of the morphology, particle sizes, and compositions of the condensed combustion products (CCP) of modified double-base propellants (1,3,5-trimethylenetrinitramine (RDX) as oxidizer) on the chamber pressure (<35 MPa) and nickel inclusion have been evaluated under a practical rocket motor operation. It has been shown that higher pressure results in smaller average particle sizes of the CCPs. The CCPs of Ni-containing propellants have more diverse morphologies, including spherical particles, large layered structures, and small flakes coated on large particles depending on the pressure. The specific surface area (SSA) of CCPs is in the range of 2.49 to 3.24 m^2^ g^−1^ for propellants without nickel are less dependent on the pressure, whereas it is 1.22 to 3.81 Ni-based propellants. The C, N, O, Al, Cu, Pb, and Si are the major elements presented on the surfaces of the CCP particles of both propellants. The compositions of CCPs from Ni-propellant are much more diverse than another one, but only three or four major phases have been found for both propellants under any pressure. The metallic copper is presented in CCPs for both propellants when the chamber pressure is low. The lead salt as the catalyst has been transformed in to Pb(OH)Cl as the most common products of lead-based catalysts with pressure lower than 15 MPa. When pressure is higher than 5 MPa, the nickel-based CCPs has been found to contain one of the following crystalline phases: Pb_2_Ni(NO_2_)_6_, (NH_4_)_2_Ni(SO_4_)_2_·6H_2_O, C_2_H_2_NiO_4_·2H_2_O, and NiO, depending on the pressure.

## 1. Introduction

In spite of great advancements in the field of new energetic materials (EMs), the 1,3,5-trimethylenetrinitramine (RDX) and ammonium perchlorate (AP) are still the most widely used ingredients as oxidizers in solid propellants [1]. In order to improve the combustion efficiency of solid propellants, one of the most effective ways is to add nano-sized catalysts. It has been shown that the catalysts could largely increase the burn rate and combustion efficiency of the solid propellants by changing the solid-state and gas-phase reaction mechanisms of between the oxidizers and fuels generated by the main ingredients such as binders, RDX, and AP. The effects of catalysts on the decomposition kinetics, reaction mechanisms, and burning rates, combustion characteristics have been widely investigated in the past decades. It has been shown that in case of nitramine-based propellants, the reaction of CH_2_O + NO_2_ → CO + NO + H_2_O is proposed to be the most important one in their foam layer that determines the burn rate, where the evaporation and condensation of nitramines and nitric esters as the binder are responsible for the dominant mass transfer [2]. In the presence of extra oxidizer, such as potassium chlorate, the major gaseous decomposition products of RDX could be changed to CO, CO_2_, HCN, NO_2_, and H_2_O due to instant consumption of CH_2_O through the gas-phase transformation CH_2_O + O_2_ → HCO +HO_2_, resulting in higher reaction rate [3].

As one of the important groups of solid propellants, nitramine-containing modified double-base propellants are featured with low-emission (sometime smokeless), high mechanical strength, long shelf-life, and very low-pressure exponents. They are widely used in rocket motors of tactical missiles, with the higher burning rates (e.g., over 25 mm s^−1^ at 7 MPa) and low-pressure exponents (n < 0.2) [4]. In order to improve the energy content of this type of propellants, usually maximum 5wt% metal fuels (e.g., Al, Mg, B, and Ni) would be included [5]. The pressure exponent could be further decreased with extended pressure range by use of novel multi-functional catalysts [6]. The experiment phenomenon on combustion performance improvement has been widely reported by use of metal powder and novel catalysts as additives. However, the inherent mechanisms of these changes are still not so clear, even after much effort has been made on combustion mechanisms and the kinetic modeling of these processes [7]. In particular, some models have been developed to predict the comprehensive properties of the coarse condensed combustion products (CCPs) as a function of propellant formulation, burning conditions, agglomeration regularities, and geometric configuration of computational region [8,9]. Recent models on AP/HTPB propellants shows that under the condition of initial combustion pressure of 3.5 MPa and a pressure reduction rate of 1000 MPa/s, a narrow diffusion chemical reaction zone could be formed in the initial stage of depressurization. Further, the diffusion and premixed dual flame appears when the pressure drops to about 1.7 MPa [10]. For the same AP/HTPB system, three different reaction mechanisms have been proposed and simulated, which are based on the global chemistry using symbolic species for the three flames, a 12-species mechanism, and a 72-step reaction mechanism with 39-species, respectively [11]. Except for the numerical simulations, the chemical structure of the propellant flame could be predicted by ReaxFF reactive force field molecular dynamics and equilibrium thermodynamics simulations [12]. It has been found that the ReaxFF agrees considerably better with experimental results for minor species than the thermodynamic simulations.

It is widely accepted that the major combustion gaseous products from nitro-based propellants are H_2_O, CO, CO_2_, H_2_, and N_2_, whereas the HCN, NH_3_, CH_4_, nitrogen oxides, benzene, acrylonitrile, toluene, furan, aromatic amines, benzopyrene, and various polycyclic aromatic hydrocarbons are detected in minor concentrations. However, the dark zone of the propellants has much different chemicals as the intermediates with higher molecular weight than those of the luminous flame zone [13]. The gas phase reactions could become more complicated if new energetic ingredients are included. For instance, in the case of metal hydrides, the ZrH_2_ was found to tune the decomposition behaviors of AP by enhancing the generation of NO in the high-temperature decomposition stage. The hydrogen released from ZrH_2_ may promote the combustion reactions in gaseous phases and therefore induce the two-stage combustion behaviors of the corresponding propellants [14]. Furthermore, the catalysts play the key role in both condensed and gas phase reactions but with much more uncertainty; even the solid-state catalytic decomposition behavior could be well characterized. The commonly used and evaluated catalysts include nano metric metal oxides [15] and ferrocene derivatives [16]. It has been proposed that both iron oxide and copper chromite are primarily acting on the condensed phase, whereas the ferrocene first acts simply as a highly reactive fuel. The resulted ferric oxide from ferrocene in the condensed phase would further catalyze the gas phase reactions [17].

In order for a complete description of the combustion mechanisms, the CCPs have to be clarified with more details. It is much more difficult to detect the intermediates of CCP in the combustion processes of solid propellants, especially under the high temperature/pressure rocket motor operation conditions due to very limited diagnosis techniques. However, the CCPs could be finally analyzed after quenching. The CCPs usually refer to the solid products of aluminized [18] or boron-containing [19] propellants, which are usually in micron sizes after agglomeration, if using the nanosized Al [20]. It has been found that a particle-laden flame zone with a sensibly reduced particle size is disclosed in the case of nanometer Al. To better understand the metal particles burning process, the reactions of a suspension of solid particles in a rapidly-heated oxidizing gas have been investigated. It has been shown that there are two reaction-onset mechanisms, which leads to a nontrivial dependence of the total reaction time on the particle size and solid-fuel concentration within the suspension [21]. Except for the experimental setups, various new diagnosis techniques have been developed to monitor the combustion process of the solid propellants. For instance, the time-resolved synchrotron X-ray imaging to view the in-situ formed aluminum agglomerates at corresponding rocket chamber pressures [22]. This technique provides real time critical data for understanding the combustion behavior of aluminized solid propellants in real rocket motors. Secondly, the gas phase species of the propellants could be experimental determined by time-of-flight mass spectrometry (ToFMS) to verify theoretical results from the density functional theory (DFT) calculations [23].

Even so much abovementioned achievement has been made in term of diagnosis and modeling of the solid propellant combustion, there is still a huge challenge to clarify the detailed reaction processes in the flame zone. It is essential to evaluate the combustion mechanisms under the operation conditions of a real rocket motor, based on which the interior ballistic characteristics could be well predicted. Owing to the extremely large specific surface areas, nano-sized catalysts have significant catalytic effects in both condensed and gas phases during decomposition and subsequent combustion, via activation of the reactants and acceleration of their transition state formations. In order to determine the relationship between the compositions, the chemical structure of the CCPs of modified double-base (MDB) propellants with different additives under a wide range rocket motor flow conditions are investigated, using a recently assembled facility with the capacity to capture almost all of the CCPs. The results may also give some evidence on how the pressure change the chemical structure of the CCPs, which can help setup the experiments for the target structure from flame synthesis under high pressure.

## 2. Experimental Procedure

### 2.1. Preparation of the Propellant Samples and Nomenclature

The MDB propellant samples were prepared by a cast-curing technique at a temperature of 70 °C for three days. The detailed ingredients of the slurry for the first typical formulation are as follows: Nitrocellulose (24.0 wt%), Nitroglycerine (30 wt%), RDX (35.5 wt%), Al_2_O_3_ (2.0 wt%), Nickel (5.0 wt%), and Lead/Copper salicylates (3.5 wt%) as the catalysts. This sample is named as “LZ” in the following sections, where the ending number means the average maximum pressure in the rocket motor. For the second typical formulation are as follows: Nitrocellulose (24.0 wt%), Nitroglycerine (30.0 wt%), RDX (40.5 wt%), Al_2_O_3_ (2.0 wt%), and Lead/Copper salicylates (3.5 wt%). This sample is named as “JZ” in the following sections, where the ending number means the average maximum pressure in the rocket motor. The burn rate law for LZ and JZ propellants are *u* = 8.36*P*^0.422^ (*P* = 1.0–10.0 MPa) and *u* = 17.67*P*^0.201^ (*P* = 10–22 MPa), respectively.

### 2.2. Rocket Motor Assembling and Combustion Condensed Products Collection

Considering that the flame temperature of the MDB propellants is between 2500–3500 K, but the ingredients of the condensed products must be analyzed at ambient conditions (about 293 K), the CCPs have to be cooled down before all structure determinations. First, the reactions between the gaseous products with the condensed products need to be prevented during the cooling process. A facility has been designed in our lab to collect the CCPs and a schematic setup is shown in Figure 1 [24].

Air was first discharged from the collection tank using a vacuum pump and then the tank was filled with argon several times (no less than three times) to prevent reactions between the combustion products and oxygen from air. The propellant grain loaded in the solid rocket motor was ignited and the combustion products were ejected into the collection tank through a nozzle. After the combustion products were cooled down naturally and the CCPs were settled, which usually takes 2 h, the door of collection tank can be opened for the sampling process. The combustion chamber pressure can be changed by adjusting nozzle throat diameter and the chamber pressure was measured by the pressure transducer during ignition and combustion processes. The CCPs were dried at 80 °C under vacuum for 12 h before analysis.

The combustion conditions of the propellants in this facility were almost the same as those in the solid rocket motor and the CCPs were widely dispersed in the stainless-steel collection tank with a large volume (about 2 m^3^), decreasing the probability of chemical reactions among the condensed-phase products to a certain extent. Large wall heat transfer occurs after ejection due to the high thermal conductivity of stainless-steel, which is also beneficial with respect to cooling the products. In addition, the collection tank was filled with inert gas (argon) and this also played an important role in avoiding reactions among the combustion products by increasing the cooling rate and diluting the combustion reactants.

It should be noted that some gaseous products ejected from the solid rocket motor, such as H_2_O may exist in the form of liquid or solid due to physical and chemical interactions during the cooling of the combustion products. The results of chemical analysis are reliable and can represent the real compositions of CCPs. Chamber pressure is important with respect to the compositions of the combustion products of the propellants. The experimental conditions, chamber pressures, and combustion time for the experiments of LZ and JZ formulations are shown in Table 1. In addition, the curves of chamber pressure vs. function time of two different MDB propellants are plotted in Figure 2.

Figure 2 shows that the chamber pressure cannot maintain constant during combustion of the propellants, which is common in the case of small rocket motors with very short working times, and a proper grain selection can help in getting neutral burning only in the case of full-scale rocket motors [25]. In order to investigate the effect of pressure on the combustion products, the time-averaged pressure *P*_c,av_ was used to characterize operation one. The pressure curve of the rocket motor could usually be divided into three parts, i.e., start-up phase, steady-state phase, and tail-off phase.

The steady-state phase clearly dominates the overall performance of the motor and the chamber pressure is usually calculated from this phase. Therefore, the starting point and end point of this phase must be determined. Figure 3 shows how to determine the function time of a solid rocket motor.

In these experiments, the starting point of the steady-state phase, *t*_1_, is defined as the time corresponding to the point of 20% of the maximum pressure on P-t curves; *t*_2_, corresponds to the transition point of the steady state to pressure drop. The time-averaged chamber pressure *P*_c,av_ could be defined as follows:$$P_{c,av}=\frac{\int_{t_{1}}^{t_{2}}Pdt}{t_{2}-t_{1}}$$

It is clear that with the increase of the chamber pressure, the function time becomes shorter due to increased burn rate. The curves become less flat, especially for the one that has peak pressure of 35 MPa, which is unexpectedly out of the calibrated scale (25 MPa) of the pressure sensor.

## 3. Results and Discussion

### 3.1. The Surface Morphologies of the Overall CCPs

As the first step, the obtained CCPs are divided into three parts based on the particle sizes. The residues with centimeter sizes were picked out by naked eyes (Appendix A), which belongs to the exfoliated or fragmented linear layer of the propellant charges due to pressure fluctuation and ablation. The particles in micron sizes are collected by sedimentation in ethanol, whereas the upper suspended part containing nano-sized particles are subject to the Transmission Electronic Microscopy (TEM) analyses in a later section. The morphologies of the micron-sized CCPs are investigated by the Scanning Electronic Microscopy (SEM) technique, which are shown in Figure 4 and Figure 5.

As shown in Figure 4a–c, there are a large amount of spherical particles with diameters of 1–3 μm for JZ propellant, which does not contain Ni metal particles in comparison to LZ propellant. It is clear with the pressure increase, the average particle size decreases and also one could notice that once the pressure achieves 35 MPa, which is out of the range of the calibrated pressure sensor. The particles morphology seems return to be similar as the one obtained at lower pressure. It should be noted that the pressure of 35 MPa is the peak pressure that has very short duration as shown in Figure 2. In this case, the pressure attenuation is dramatically after the peak value, so that the CCPs formed are similar to those obtained under lower pressure. It is shown in Figure 4a_1_–c_1_ that there are a large amount of amorphous nano-particles deposited on those large spherical particles. The elemental analyses show that these materials in most cases are nanocrystalline inorganic residues including carbon soot particles.

Similarly, Figure 5 presents the surface structures of CCPs from LZ propellant at three different pressures as a comparison. With metallic Ni included, the LZ propellant has very different CCPs with a variety of morphologies, and the most typical ones are spherical particles, large layered structures, and small flakes coated on large particles. In general, the particle size of the spherical CCPs from LZ propellant is larger than those of JZ propellant. In order to clarify what elements are included in the CCPs, the EDS analyses have been done, with the spectra plotted in Appendix A, whereas the detailed elemental analyses are shown in Appendix A, and the average atomic contents are summarized in Table 2.

It is clear from Table 2 that, the C, N, O, Al, Cu, Pb, and Si are the major elements presented on the surfaces the CCP particles. For JZ propellants, with the increase of pressure, the content of carbon increases, and the O and Si decreases. The silicon element comes from the insulating material, which is based on a silicon rubber, indicating erosive burning to the insulation layer. However, higher pressure seems less erosive to the insulation layer for the JZ propellant free of metal fuel like nickel. It is opposite when the 5% of nickel is included as the metal fuel and flame stabilizer, where the erosive burning increases with the pressure, resulting in much higher Si content in the CCPs, e.g., it is 3.52% for LZ-18 and 3.06% for JZ-20, respectively. The organic copper and lead salts are mainly transformed into CCPs. However, when the pressure is very high for the JZ propellant, there is no interaction between catalysts with the metal fuel. The content of Cu would drop to very low as 1.96%, which is opposite of the LZ propellant. As shown in a later section, based phase analyses by X-ray Diffraction (XRD) show the higher pressure results in higher content of Cu and Pb in the CCPs, due to formation of some less volatile products such as minerals or alloys under the effect of nickel. The other minor elements such as Ca, K, and Cl are mainly from the complicated insulating materials. In terms of Al element, which comes from Al_2_O_3_ as a ballistic stabilizer, the content first increases and then decreases with the pressure for JZ propellant, whereas it largely increases with the pressure to even 11.8% when the peak pressure is 35 MPa for LZ propellant. The interaction between the Al_2_O_3_ with metallic nickel is significant and the higher pressure is favorable for such mutual reactions, resulting in more nonvolatile products, which is responsible for such an increase in Al element content.

### 3.2. The Particle Size Distributions of the Micron-sized CCPs

Based on the separation of the nano-sized and micron-sized CCP particles by sedimentation method in aqueous media, the micron-sized particles are subject to a size distribution analysis. The corresponding distribution curves are plotted in Appendix A, with the detailed parameters for these plots are summarized in Appendix A. In order for better comparison of the average values of the distribution based on repeated experiments, the normalized curves are shown in Figure 6 and Figure 7 for JZ and LZ propellants, respectively. It is clear from Figure 6 that the micron-sized CCPs from combustion of JZ propellants can be divided into three groups in terms of particle sizes: submicron ones, media ones (1–10 μm) and larger ones (>10 μm). Most of the particles have the diameters less than 80 μm for the JZ propellant burning at any pressure.

When the pressure is less than 12 MPa, the CCPs with submicron sizes are not well separated with the media sizes, as well as the larger sizes. It means that the size boundary is not so clear under lower pressure burning of JZ propellant. In comparison, when the pressure is higher than 15 MPa, the three peaks for three groups of particle sizes become more independent with more obvious boundaries.

In terms of LZ propellants, the distributions are much different, which is highly dependent on the pressure. When the pressure is lower than 9 MPa, there are three major groups as the JZ propellant: micron and submicron (<5 μm), media sizes (5–15 μm), and large sizes (>15 μm). However, we can see that there is a strange narrow peak and large shoulder peak at the size over 200 μm at the pressure of ambient pressure and 2 MPa, which is probably due to presence of large agglomerated Nickel, which is too large to do the SEM analyses. It has been mentioned that the content of Al is largely dependent on the pressure and under lower pressure there is less interaction between Al and Ni in LZ propellant, resulting in lower burning efficiency and larger agglomeration.

When the pressure is over 18 MPa, the CCP from burning of LZ propellants, the amount of micron and submicron particles is greatly increased. Thus, the overall peak becomes narrower due to less agglomeration of the Nickel particles. For better quantitative comparison of the particle sizes, the detailed parameters are summarized in Table 3. It is clear that the specific surface area (SSA) of CCPs from the JZ propellant is less dependent on the pressure, which is in the range of 2.49 to 3.24 m^2^ g^−1^. It is 1.22 to 3.81 for the LZ propellant due to high pressure dependence of metallic nickel transformation and interaction of elemental Ni with Al_2_O_3_. The high pressure is favorable for more complete transformation and lower particle sizes [26]. The presence of less volatile products would decrease the size of CCP and, for instance, the formation of AlF_3_ would greatly mitigate the agglomeration of Al combustion by using organic fluorine-contained additives [27].

In this case, the polymorphic transition of the metal oxide is also possible, e.g., α-Al_2_O_3_ and θ-Al_2_O_3_ may be transformed into δ-Al_2_O_3_ and γ-Al_2_O_3_. In the presence of Al_2_O_3_ in LZ and JZ propellants as a stabilizer, it may react with carbon soot and many other organic residues to form various other condensed products instead of polymorphs of Al_2_O_3_. The cooling mode may also affect the particle size and phases content of the CCPs, but the comparative study under the same experimental conditions is fine [28].

### 3.3. The Interface Structure of the Nanosized CCPs

In order to observe the interface structures and the shape of the nano-sized CCPs, the TEM images of selected samples are collected and shown in Figure 5 and Figure 6 for JZ and LZ propellants, respectively. The overall pictures of the nano-sized CCPs from combustion of JZ propellant under the pressure of 7, 12, and 15 MPa are shown in Figure 8a–c. The maximum sizes of the CCPs increases with the pressure and then decreases with the shape changes. The particle shapes become less uniform when the pressure increases. Generally, the spherical particles with diameters in the range of 200–400 nm could be formed under the pressure of 7 MPa. There are some needles and nano-flakes can be observed when the pressure is over 12 MPa, indicating new crystal phases. Large irregular aggregates could be found when the pressure achieves 25 MPa, where the spherical particle has a diameter of about 200–300 nm.

In comparison, as shown in Figure 9a–c, the maximum sizes of the nano-sized CCPs increases with the pressure and the morphology changes as well. There are generally three types of particle shapes: stacked layers, spherical, and long needle-like. Under the ambient pressure, there are a large amount of small particles with diameter of less than 100 nm, which are deposited on stack-layered structures. When the pressure increases to 7 MPa, the layers become thicker and the needles appears with length over 500 nm and diameter less than 50 nm. With the pressure further increases, the length and diameter of the needle structure decreases (Figure 9c1). Generally, for LZ propellant, the spherical particles with diameters in the range of 50–100 nm under lower pressure, but it could be over 500 nm under the pressure of 35 MPa. Large irregular aggregates could be stick to the spherical particles at this high pressure. It has been proposed for metallized propellants that the irregular agglomerates usually larger than spherical agglomerates [29,30]. The formation of irregular agglomerates was proposed to occur by three steps: deformation of spherical metal drops, combination of particles with various shapes, and formation of irregular aggregations.

### 3.4. The Chemical Compositions of the CCPs

As the most important step, the determination of the crystal phases inside the CCPs has to be done, to show the inherent reaction mechanism in the gas phase under a real rocket motor flow condition. The powder X-ray (PXRD) spectra have been collected, with the crystal phases identified in Figure 10 and Figure 11. The quantitative analyses of the content of the selected major phases are listed in Appendix A.

It has been shown from Figure 10 and Figure 11 that the compositions of the crystal phases of CCPs are highly dependent on the pressure. In general, the CCPs from the LZ propellant are much more diverse than those from the JZ propellant. In the case of the JZ propellant under the attempted pressures, only three to four major phases have been identified under each pressure. The metallic copper is shown when the pressure is not higher than 20 MPa, the content slight decreases with the increase of pressure. It means the organic copper salts would transform to CuO first, which further participate in the reaction with carbon soot as an oxidizer. The presences of AlCu_3_ at the pressure of 12 and 35 MPa and AlCu_4_ at the pressure of 15 MPa indicate interaction of the resulted metallic Cu with the Al_2_O_3_. Only 13% of Al_2_O_3_ is found in CCP of JZ under pressure of 20 MPa, which means the reaction between Al_2_O_3_ with the other products in the gas phase is not complete at this pressure. It may be transformed to Al_2_O at lower pressure (e.g., <7 MPa).

The lead salt as the catalyst has been transformed in to Pb(OH)Cl at the pressure lower than 15 MPa or Pb_2_Cl_2_(CO_3_) when the pressure is higher than 20 MPa. The Pb_2_Cl_2_(CO_3_) is also so-called phosgenite as a naturally formed mineral, which are usually colorless and transparent. This mineral is rather sectile and consequently was earlier known as corneous lead. This finding may help in designing novel preparation method for this mineral by a flame synthesis process using lead-containing propellant as a precursor. Besides, a large amount of carbon soot was formed when the pressure achieves 35 MPa, due to extinguish of the CCP intermediates caused by fast pressure drop after the peak pressure (see in Figure 2). The unique product SiO_2_ from JZ-12 is due to the contaminant of the exhausted insulating materials after oxidation.

In comparison, as shown in Figure 11 and Appendix A, it is similar to LZ propellants, only three or four major crystalline components have been found under each pressure. The Pb(OH)Cl is the most common product of lead salts as the catalysts, whereas the copper salt could be transformed into various compounds depending on the pressure, e.g., it is CuO under ambient pressure. It is logical, then, that the gas phase reaction was not realized under the ambient pressure and the CuO as the condensed decomposition product of organic cooper salt did not change. The Cu_3_N_1_, C_12_H_27_N_3_O_6_·CuC_l2_·2H_2_O, Cu_1.8_S and Cu(NO_3_)_2_·3H_2_O are unique copper based compounds under the pressure of 7, 9, 18, and 35 MPa. The lead salt would transform to Pb_2_SO_5_, 2Pb(CO_3_)·Pb(OH)_2_ and Pb_4_O_3_Cl_2_·H_2_O at the pressure of 7, 12, and 18 MPa. At the pressure of 5 MPa, the elemental lead even combines with nickel to form Pb_2_Ni(NO_2_)_6_. It is interesting that at the same pressure of 12–15 MPa, no matter the LZ or JZ propellant, the copper aluminum alloy could be formed. It is AlCu_4_ for the LZ propellant and AlCu_3_ or AlCu_4_ for the JZ propellant.

As mentioned earlier in this paper, the presence of metallic nickel in the formulation makes the compositions of CCPs largely dependent on the pressure. The diverse of the crystalline components of CCPs derives mostly from the nickel element. Under ambient pressure, the metallic nickel is presented due to incomplete combustion, and the other nickel-based compounds are not shown. With the increase of pressure (>5 MPa), the nickel-based CCPs are Pb_2_Ni(NO_2_)_6_, (NH_4_)_2_Ni(SO_4_)_2_·6H_2_O, C_2_H_2_NiO_4_·2H_2_O, and NiO. At the pressure of 9 MPa, the CCP contains Si-based compound HAlO_10_Si_4_, which is different from the JZ propellant (SiO_2_). One could notice above that the high dependence of gas phase reactions on pressure, which results in a variety of CCPs in terms of different compositions and morphologies. In terms of Boron-based metal fuels, such as B-Mg and B-Al, the CCPs with ferrocene as the catalyst could be even more complicated. It has been shown that the condensed-phase products are mainly composed of B, C, B_4_C (or B_12_C_2_), BN, Mg, MgO, MgAl_2_O_4_, Al, Al_2_O_3_, AlCl_3_, NH_4_[Mg(H_2_O)_6_]Cl_3_, NH_4_Cl, and Fe_3_O_4_ [24]. It was shown that the high pressure is not so benefit to high efficiency secondary combustion of boron in gas phase due to formation of less active boron carbide, graphite, and h-BN. As a comparison, the CCPs from aluminum boride and boron–aluminum blends are very similar, with the common ingredients of B, Al, B_2_O_3_, Al_2_O_3_, AlN, and Al_5_O_6_N, showing incomplete combustion due to low pressure [31]. In our case, the high pressure is favorable for secondary reaction of Ni-based compounds and the NiO is the major product under higher pressure, without metallic nickel detected.

## 4. Conclusions

In this paper, the dependences of the morphology, particle sizes, and compositions of the CCPs of modified double-base propellants on the chamber pressure and nickel inclusion have been evaluated under conditions for a real rocket motor operation. The following conclusions could be made:(1)The peak chamber pressure has been successfully modulated by changing the size of the throat of the motor. The maximum pressure achieved herein is about 35 MPa, and the lowest one is the ambient pressure.(2)With the pressure increase, the average particle size of the CCPs decreases. In presence of metallic Ni, the CCPs have a variety of morphologies depending on the pressure. The most typical ones are spherical particles, large layered structures, and small flakes coated on large particles. The SSA of CCPs from the JZ propellant is less dependent on the pressure, which is in the range of 2.49 to 3.24 m^2^ g^−1^, whereas it is 1.22 to 3.81 for the LZ propellant due to high pressure dependence of metallic nickel transformation and interaction of elemental Ni with Al_2_O_3_.(3)The C, N, O, Al, Cu, Pb, and Si are the major elements presented on the surfaces the CCP particles. The micron-sized CCPs from combustion of both propellants can be divided into three groups in terms of particle sizes, but the LZ propellant containing metallic nickel has larger CCPs sizes.(4)The compositions of CCPs from LZ propellant are much more diverse than those from JZ propellant, but only three to four major phases have been found under each pressure. The metallic copper is presented in CCPs for both propellants when the chamber pressure is low.(5)The lead salt as the catalyst has been transformed in to Pb(OH)Cl as the most common products of lead-based catalysts at the pressure lower than 15 MPa or Pb_2_Cl_2_(CO_3_) when the pressure is higher than 20 MPa. The copper salt and nickel could be transformed into various compounds depending on the pressure and when pressure is greater than 5 MPa, the nickel-based CCPs contain one of the following phases: Pb_2_Ni(NO_2_)_6_, (NH_4_)_2_Ni(SO_4_)_2_·6H_2_O, C_2_H_2_NiO_4_·2H_2_O, and NiO.

## Figures and Tables

**Figure 1 nanomaterials-09-00381-f001:**
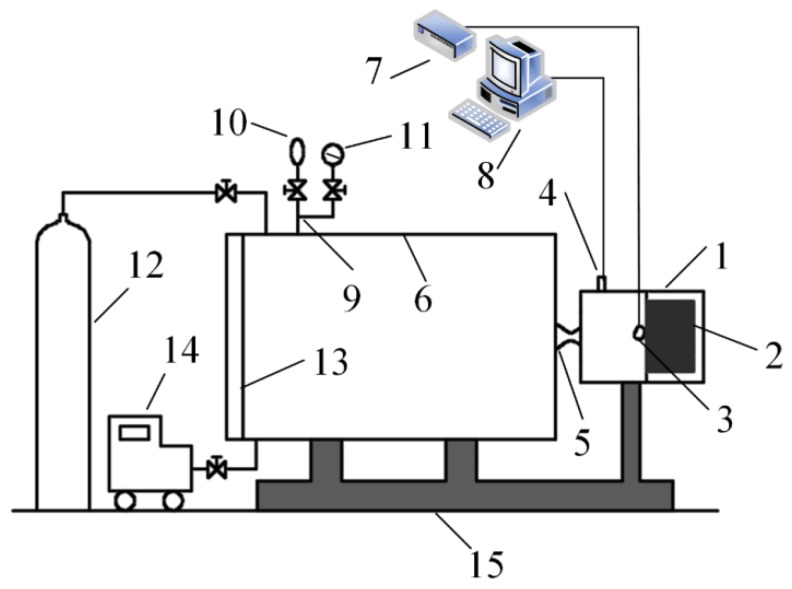
Schematic of the experimental facility: (1) solid rocket motor; (2) propellant grain; (3) igniter pad; (4) pressure transducer; (5) Laval nozzle; (6) collection tank; (7) ignition controller; (8) signal acquisition system; (9) gaseous product collecting piping/gas exhaust piping; (10) Teflon bag; (11) vacuum manometer; (12) hyperbaric argon cylinder; (13) door of collection tank; (14) vacuum pump; (15) steel holder and stabilizer; the shape of propellant grain, solid cylindrical coated with silicone rubber based heat insulation layer.

**Figure 2 nanomaterials-09-00381-f002:**
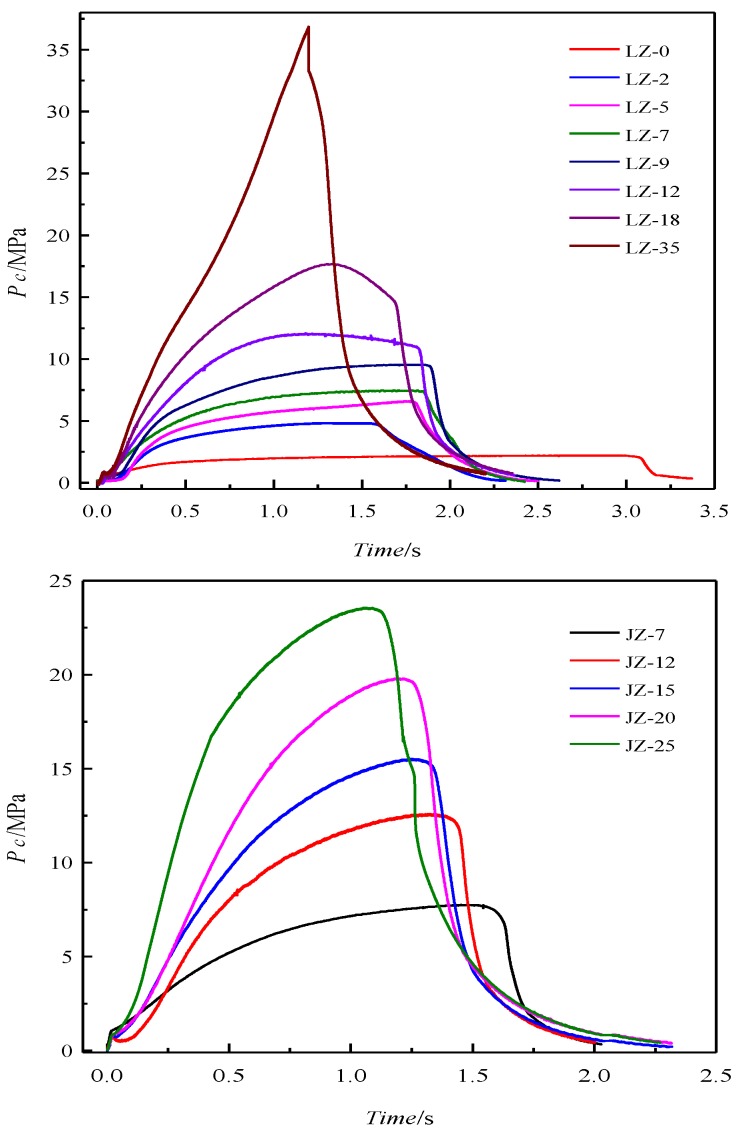
The pressure–time curves in the experimental rocket chamber for all firing measurements.

**Figure 3 nanomaterials-09-00381-f003:**
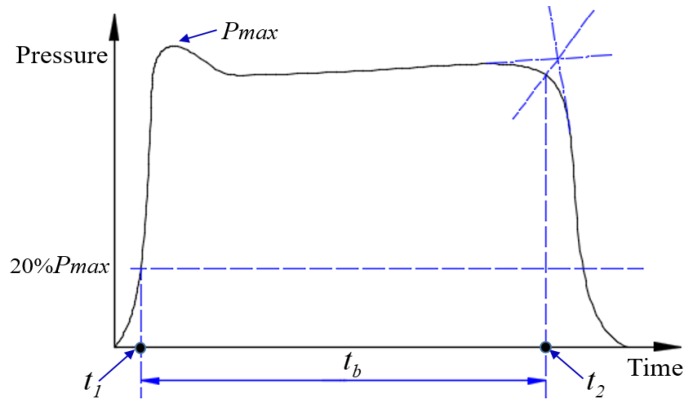
Function time diagram of solid rocket motor.

**Figure 4 nanomaterials-09-00381-f004:**
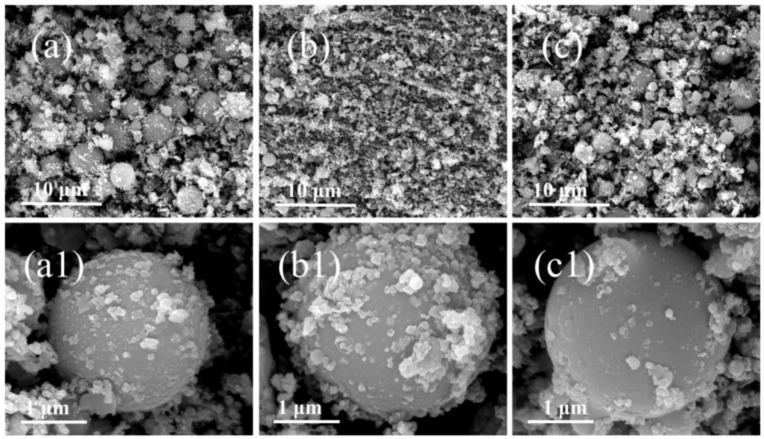
The SEM images of combustion products of JZ propellant grains: (**a**,**a_1_**) JZ-15; (**b**,**b_1_**) JZ-20; and (**c**,**c_1_**) JZ-35.

**Figure 5 nanomaterials-09-00381-f005:**
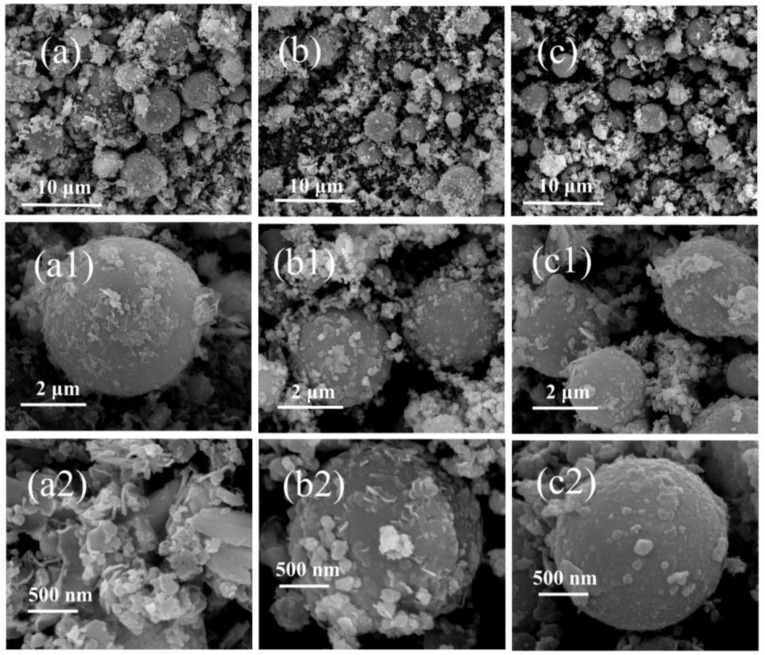
The SEM images of CCPs from LZ propellant (**a**,**a_1_**,**a_2_**) LZ-7; (**b**,**b_1_**,**b_2_**) LZ-18; (**c**,**c_1_**,**c_2_**) LZ-35.

**Figure 6 nanomaterials-09-00381-f006:**
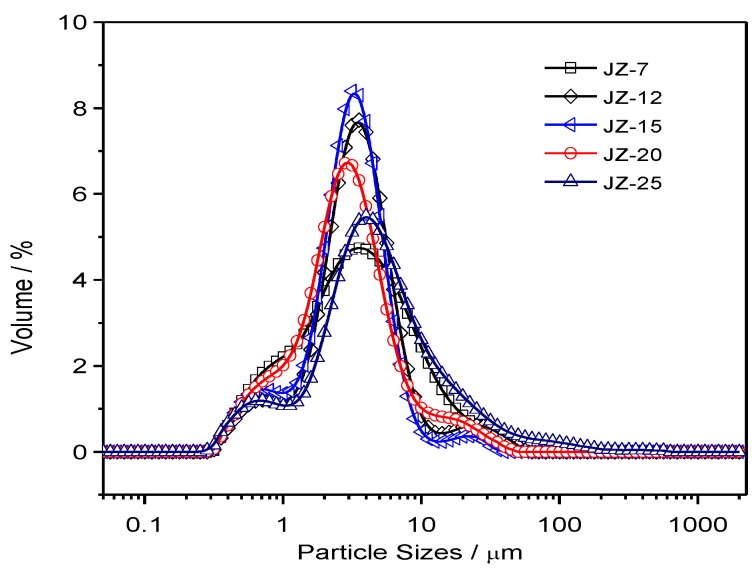
The particle size distributions of CCPs from combustion of JZ propellant grains under various pressures.

**Figure 7 nanomaterials-09-00381-f007:**
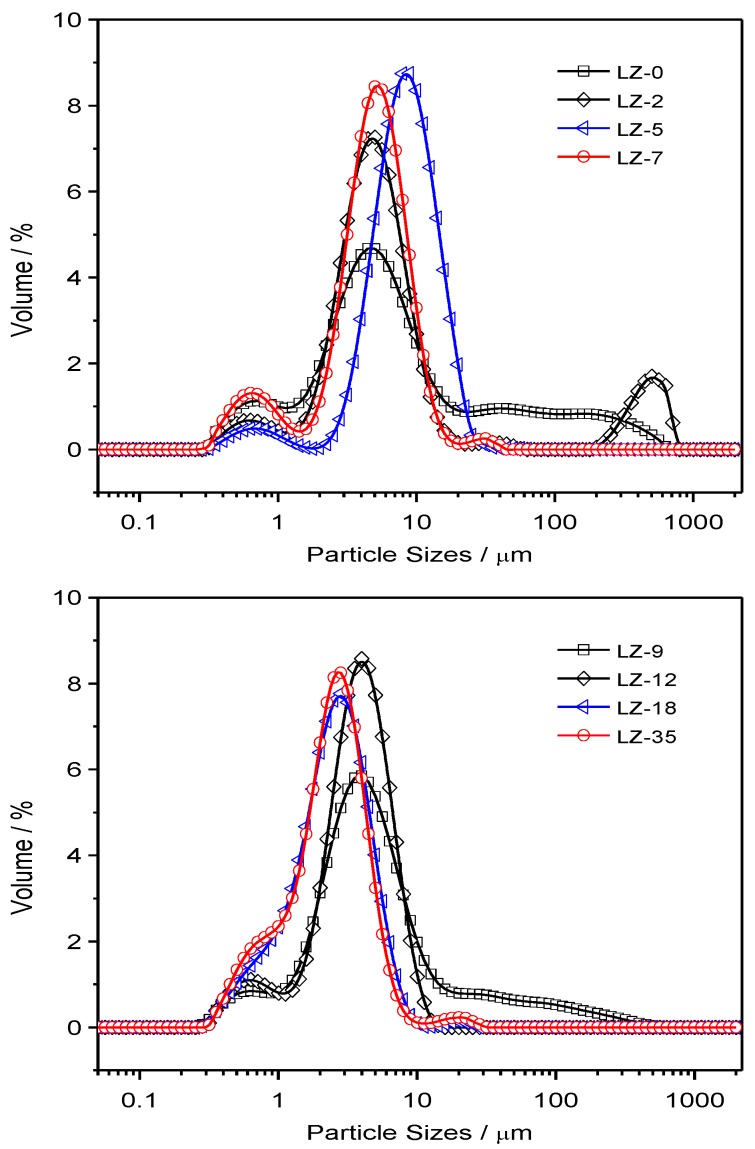
The particle size distributions of condensed combustion products (CCP) from combustion of LZ propellant grains under various pressures.

**Figure 8 nanomaterials-09-00381-f008:**
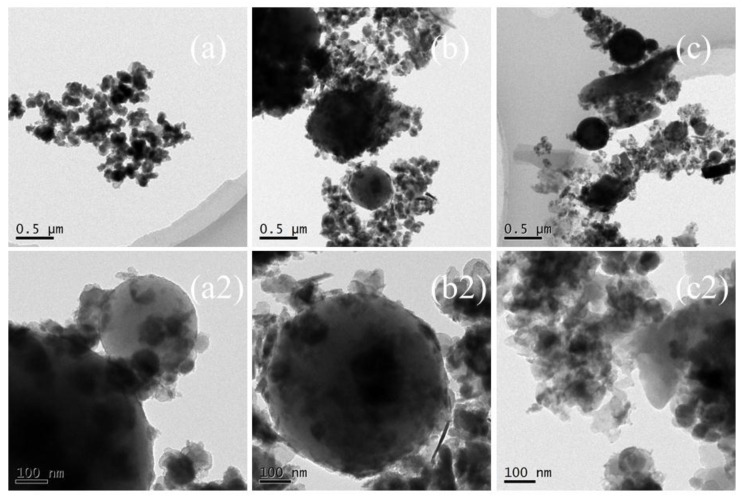
The TEM images of nano-sized CCPs of JZ propellant grains (**a**,**a_2_**) JZ-7; (**b**,**b_2_**) JZ-12; (**c**,**c_2_**) JZ-25.

**Figure 9 nanomaterials-09-00381-f009:**
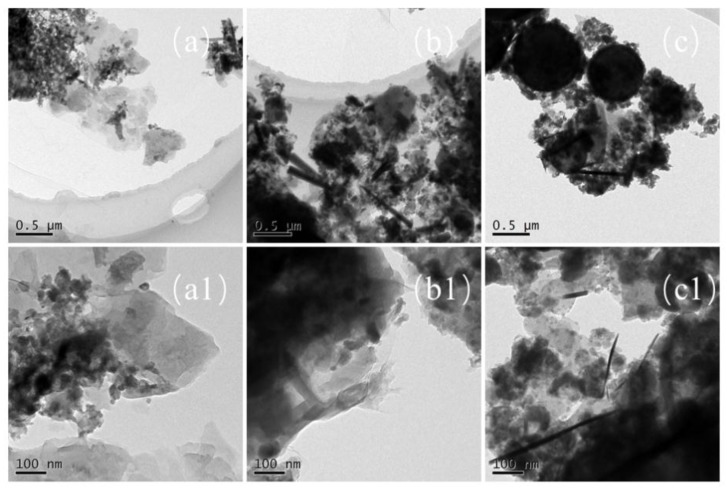
The TEM images of nano-sized CCPs of LZ propellant grains (**a**,**a_1_**) LZ-0; (**b**,**b_1_**) LZ-7; (**c**,**c_1_**) LZ-35.

**Figure 10 nanomaterials-09-00381-f010:**
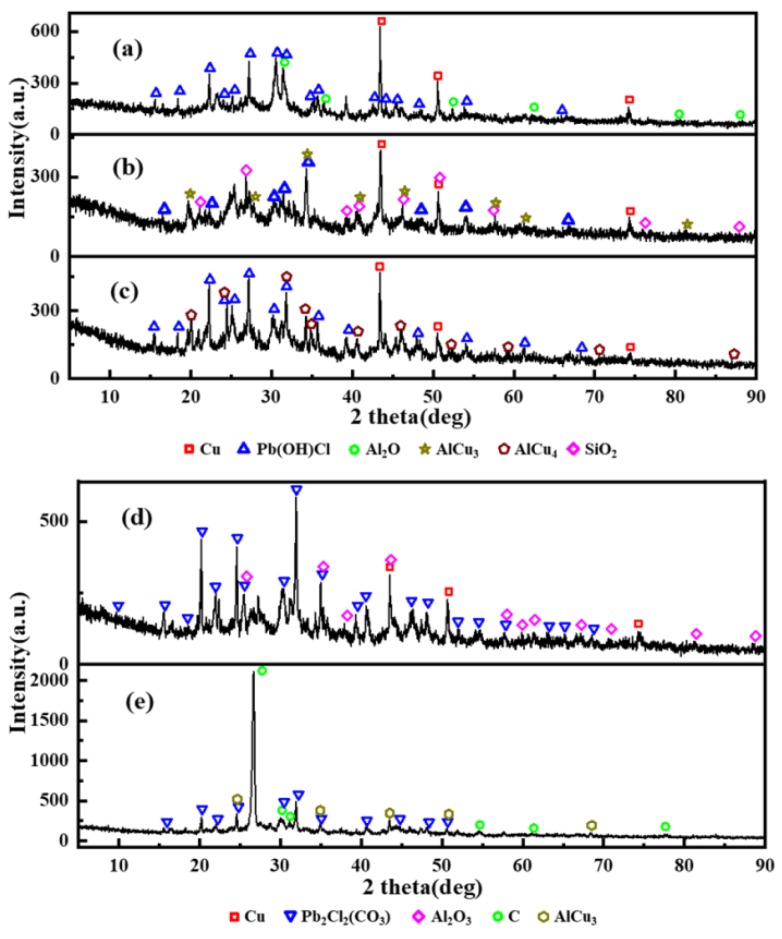
X-ray diffraction (XRD) results of JZ propellant grains combustion product; (**a**–**e**) represent pressure of 7, 12, 15, 20, and 25 MPa).

**Figure 11 nanomaterials-09-00381-f011:**
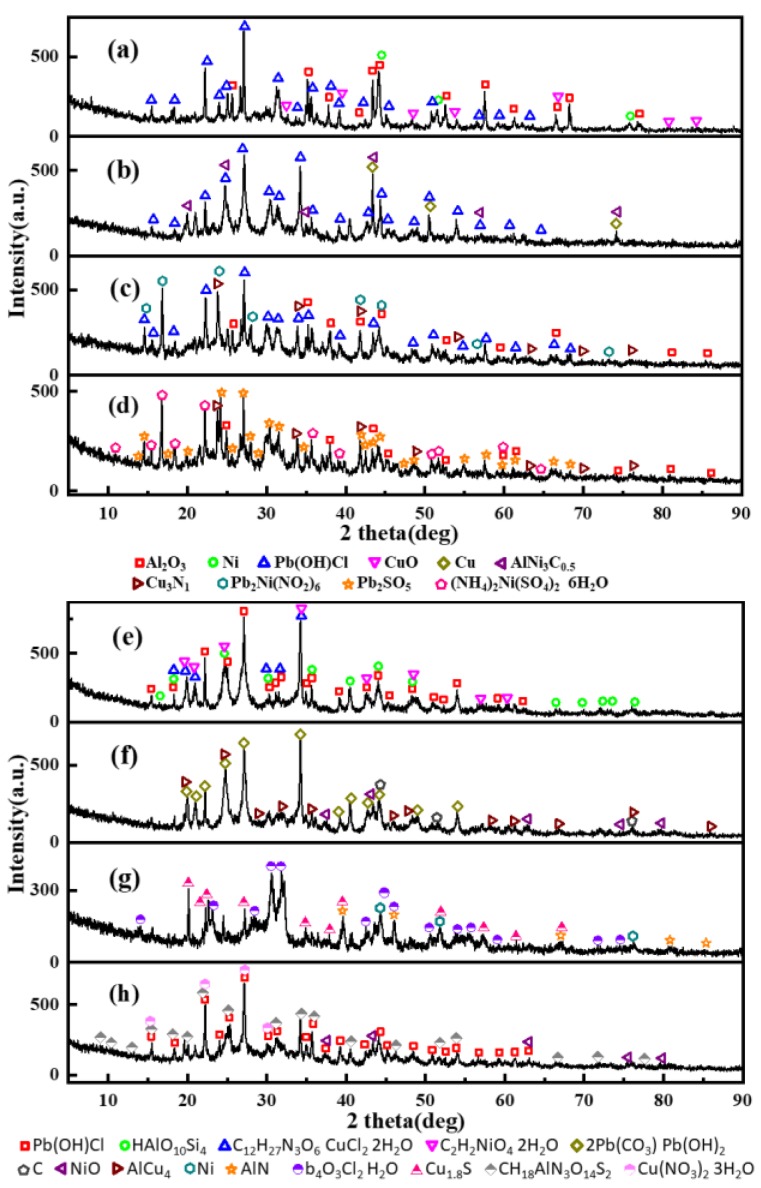
XRD results for condensed combustion products of LZ propellants; (**a**–**h**) represent pressure of 0, 2, 5, 7, 9, 12, 18, and 35 MPa.

**Table 1 nanomaterials-09-00381-t001:** Experimental conditions and experimental results of LZ propellant grains.

Samples	*D*_t_/mm	*P*_c,max_/MPa	*P*_c,av_/MPa	*t*_b_/s
JZ-7	5.60	7.7	5.8	1.58
JZ-12	4.70	12.5	9.4	1.27
JZ-15	4.25	15.5	11.1	1.22
JZ-20	3.91	19.8	13.6	1.14
JZ-25	2.84	23.6	20.3	0.93
LZ-0	7.50	0.8	0.5	2.96
LZ-2	6.20	2.2	2.0	1.46
LZ-5	5.40	6.6	5.3	1.63
LZ-7	5.01	7.4	6.1	1.76
LZ-9	4.50	9.5	7.8	1.74
LZ-12	4.00	12.0	9.8	1.71
LZ-18	3.30	17.7	13.2	1.56
LZ-35	2.50	36.8	19.3	1.05

Notes: *D*_t_ is nozzle throat diameter. *P*_c,max_ is maximum chamber pressure of rocket motor. *P*_c,av_ is average chamber pressure of rocket motor. *t*_b_ is function time.

**Table 2 nanomaterials-09-00381-t002:** Element distribution results of JZ and LZ propellant grains combustion products.

Elements	Average Atomic Contents (%)					
	JZ-15	JZ-20	JZ-25	LZ-7	LZ-18	LZ-35
C K	30.27	31.09	44.43	40.13	42.33	26.08
N K	3.38	2.66	3.74	13.80	1.51	2.33
O K	43.74	41.84	34.92	31.27	34.60	44.02
Al K	9.55	10.18	7.60	4.16	7.50	11.80
Si K	3.21	3.06	2.31	1.51	3.52	3.41
P K	0.25	0.34	0.30	0.18	0.79	0.12
S K	1.10	1.26	0.91	-	-	-
Cl K	0.70	1.02	0.86	1.27	0.89	1.18
K K	0.68	0.62	0.8	0.99	0.60	0.70
Ca K	0.24	0.22	0.19	0.14	-	0.26
Fe K	0.99	0.93	0.42	0.66	0.86	0.53
Ni K	0.28	0.22	0.29	2.16	2.51	3.80
Cu K	4.33	5.00	1.96	1.41	1.74	2.52
Pb M	1.33	1.60	1.31	2.36	2.27	3.28

**Table 3 nanomaterials-09-00381-t003:** Particle size distributions of the CCPs for involved JZ and LZ propellants.

Samples	Obscuration	Residual	Concent.	Span	D_[4,3]_ ^a^	Uniformity	SSA	D_[3,2]_ ^b^	d_(0.1)_ ^c^	d_(0.5)_ ^d^	d_(0.9)_ ^e^
JZ-7	7.60	1.317	0.0027	3.154	4.981	1.050	3.06	1.959	0.794	3.179	10.819
JZ-12	8.90	1.144	0.0033	1.992	4.588	0.842	2.77	2.167	1.035	3.167	7.343
JZ-15	8.55	1.798	0.0029	1.685	3.483	0.606	3.08	1.945	0.875	2.876	5.721
JZ-20	9.31	1.125	0.003	2.512	3.959	0.905	3.24	1.855	0.834	2.661	7.519
JZ-25	8.46	0.623	0.0037	3.967	9.567	1.780	2.49	2.41	0.99	4.116	17.317
LZ-0	8.37	0.571	0.0044	18.859	33.566	5.830	2.13	2.818	1.084	5.298	101.003
LZ-2	10.39	0.865	0.0057	61.074	54.834	10.800	1.83	3.286	1.995	4.788	294.406
LZ-5	8.23	1.614	0.0069	1.392	8.51	0.445	1.22	4.929	3.761	7.714	14.498
LZ-7	8.92	1.985	0.0042	1.708	5.126	0.530	2.35	2.549	0.986	4.561	8.777
LZ-9	10.16	0.482	0.0047	7.373	15.425	3.230	2.29	2.619	1.308	4.06	31.245
LZ-12	9.05	0.741	0.0035	1.541	3.758	0.464	2.64	2.27	1.18	3.481	6.544
LZ-18	9.28	1.561	0.0027	1.638	2.676	0.530	3.53	1.7	0.839	2.349	4.685
LZ-35	10.00	1.736	0.0028	1.562	2.496	0.533	3.81	1.574	0.75	2.191	4.171

Notes: a, the weighted average of the particle size to the surface area; b, the weighted average of the particle size to the volume. c, particles whose diameter equals to or less than the value of D_(0.1)_, and the sum of the volume fractions of which accounts for ten percent; it is the same for d and e; SSA, specific surface area, in m^2^/g.

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
