# Peer review of "Transformation of Combustion Nanocatalysts inside Solid Rocket Motor under Various Pressures"

_nanomaterials, 2019, doi:10.3390/nano9030381_

Reviewer 1 Report

Novel and interesting paper, with good and valuable experimental results carefully collected under realistic motor operating conditions. The paper fully deserves publication.

Minor remarks to improve the manuscipt:

A bit of English polish would help.

p. 2 line 50: "pressure component" should be "pressure exponent"

p. 4 Fig 1: please specify the shape of the propellant grain

p. 5 line 171: a proper grain selection can help in getting neutral burning

p. 5 line 182: there is an extensive literature about this matter that should be mentioned; for example: 

Miller, W.H. and Barrington, D.K., “A Review of Contemporary Solid Rocket Motor Performance Prediction Techniques,” Journal of Spacecraft and Rockets, 7 (3): 225-237, 1970.

p. 6 line 192 : in ballistic studies "erosive burning" has a well-defined technical meaning associated with the propellant grain shape, which is different from that implied in this work (burning of insulating material)

p. 8 lines 228-233: this sentence is too long

Fig. 5-7: the double peak structure evident in metalized AP propellants is only hinted here; could the authors elaborate about this point?

Author Response

Novel and interesting paper, with good and valuable experimental results carefully collected under realistic motor operating conditions. The paper fully deserves publication. Minor remarks to improve the manuscript: A bit of English polish would help.

[QLY]: Additional English polishing has been done. Thanks.

p. 2 line 50: "pressure component" should be "pressure exponent"

[QLY]: Thanks. It has been corrected.

p. 4 Fig 1: please specify the shape of the propellant grain

[QLY]: The shape of the propellant grain has been specified.

p. 5 line 171: a proper grain selection can help in getting neutral burning

[QLY]: Thanks for the comments, which has been stated in the text. In our case, the combustion chamber is relatively larger than designed propellant charges. The charges are about 5 cm in diameter and 5 cm in length. If we use hollowed charge, the neutral burning is even more difficult to obtain.

p. 5 line 182: there is an extensive literature about this matter that should be mentioned; for example:  Miller, W.H. and Barrington, D.K., “A Review of Contemporary Solid Rocket Motor Performance Prediction Techniques,” Journal of Spacecraft and Rockets, 7 (3): 225-237, 1970.

[QLY]: This literature has been added to a proper location in the text.

p. 6 line 192: in ballistic studies "erosive burning" has a well-defined technical meaning associated with the propellant grain shape, which is different from that implied in this work (burning of insulating material)

[QLY]: Yes. It is true. We made a mistake, and now the word “ablation” is used for description.

p. 8 lines 228-233: this sentence is too long

[QLY]: This sentence has been reconstructed into several short sentences.

Fig. 5-7: the double peak structure evident in metalized AP propellants is only hinted here; could the authors elaborate about this point?

[QLY]: Additional comments are actually provided in the discussion section on how the double peak structure has been formed, which is due to presence of two types of ingredients: metal-based composites and inorganic composites in the CCPs.

Reviewer 2 Report

This manuscript presents a rather thorough assessment of the exhaust products (condensed combustion products) of two different solid propellant rocket formulations with catalytic additives. It presents one of the best examples I have seen of a research formulation in a practical test of an actual rocket motor (over a range of pressures), with a careful collection and analysis of the CCPs. The authors present an experiment that is well thought out and seemingly effective in collecting samples without contamination or oxidation beyond what would have occurred in the actual rocket motor. 

The paper is well presented (albeit some additional English grammar corrections are still needed), particularly in the description of the test procedures. The introduction and background are quite thorough. The results are not stellar, but the fact that they have been able to get well defined and careful results, as much as possible in such a challenging environment, makes this paper worthy of publication. The CCP samples are analyzed with particle sizing, SEM, TEM, and XRD analyses. Some important findings with regard to pressure effects and Ni loading are presented.

The Table 1 caption just mentions LZ grains, but I think it should also state that the JZ grains are summarized as well in the table. In this same table, the numbers for tb and the pressures seem rather precise. I would think that there are too many decimal places in these numbers as provided. Please check and update as needed.

When calculating the avg Pc, the authors use very specific definitions to determine the initial and final times. I suggest that they add an additional figure that shows some example pressure time histories with the t1 and t2 clearly marked, so the reader can see some examples. Also, the numbers and axis labels in Fig. 2 seem a little too small and should be increased in size. 

Author Response

This manuscript presents a rather thorough assessment of the exhaust products (condensed combustion products) of two different solid propellant rocket formulations with catalytic additives. It presents one of the best examples I have seen of a research formulation in a practical test of an actual rocket motor (over a range of pressures), with a careful collection and analysis of the CCPs. The authors present an experiment that is well thought out and seemingly effective in collecting samples without contamination or oxidation beyond what would have occurred in the actual rocket motor. The paper is well presented (albeit some additional English grammar corrections are still needed), particularly in the description of the test procedures. The introduction and background are quite thorough. The results are not stellar, but the fact that they have been able to get well defined and careful results, as much as possible in such a challenging environment, makes this paper worthy of publication. The CCP samples are analyzed with particle sizing, SEM, TEM, and XRD analyses. Some important findings with regard to pressure effects and Ni loading are presented.

[QLY]: Additional English polishing has been done. Thanks very much for the positive comments.

The Table 1 caption just mentions LZ grains, but I think it should also state that the JZ grains are summarized as well in the table. In this same table, the numbers for tb and the pressures seem rather precise. I would think that there are too many decimal places in these numbers as provided. Please check and update as needed.

[QLY]: Thanks. You are right. We have eliminated unnecessary decimal places.

When calculating the avg Pc, the authors use very specific definitions to determine the initial and final times. I suggest that they add an additional figure that shows some example pressure time histories with the t1 and t2 clearly marked, so the reader can see some examples. Also, the numbers and axis labels in Fig. 2 seem a little too small and should be increased in size.

[QLY]: In order to clarify the definition of these parameters, a new separate figure (Fig. 3) has been plotted and added to the paper. The corresponding description has also been presented. Once again, thanks for your kind suggestions.

Sincerely,

Q.-L. Yan